# Strain Energy and Entropy Based Scaling of Buckling Modes

**DOI:** 10.3390/e25121630

**Published:** 2023-12-06

**Authors:** Zdeněk Kala

**Affiliations:** Institute of Structural Mechanics, Faculty of Civil Engineering, Brno University of Technology, 602 00 Brno, Czech Republic; kala.z@fce.vutbr.cz

**Keywords:** strain energy, entropy, buckling, structural mechanics, structural engineering, steel structures, initial imperfections

## Abstract

A new utilization of entropy in the context of buckling is presented. The novel concept of connecting the strain energy and entropy for a pin-ended strut is derived. The entropy of the buckling mode is extracted through a surrogate model by decomposing the strain energy into entropy and virtual temperature. This concept rationalizes the ranking of buckling modes based on their strain energy under the assumption of given entropy. By assigning identical entropy to all buckling modes, they can be ranked according to their deformation energy. Conversely, with identical strain energy assigned to all the modes, ranking according to entropy is possible. Decreasing entropy was found to represent the scaling factors of the buckling modes that coincide with the measurement of the initial out-of-straightness imperfections in IPE160 beams. Applied to steel plane frames, scaled buckling modes can be used to model initial imperfections. It is demonstrated that the entropy (scale factor) for a given energy roughly decreases with the inverse square of the mode index. For practical engineering, this study presents the possibility of using scaled buckling modes of steel plane frames to model initial geometric imperfections. Entropy proves to be a valuable complement to strain energy in structural mechanics.

## 1. Introduction

The investigation of buckling modes is of great importance in the field of structural engineering and structural mechanics. Understanding the stability behaviour of structures under various loads and conditions is essential to ensuring their safety and reliability [1,2].

Throughout history, the study of structural stability has been intertwined with the evolution of engineering and mathematics [3]. The concept of buckling, where slender columns or beams, which are subjected to axial loads, deform and potentially fail under compression, has captured the attention of engineers and mathematicians for centuries [4]. The history of buckling analysis can be traced back to the 18th century in the works of Euler, who has made significant contributions to the understanding of the stability of columns [5].

As theory advances, so does our ability to investigate the structural behaviour of frame structures [6,7]. The emergence of computer-based analysis and numerical methods have enabled engineers to explore the intricacies of buckling modes in various configurations. Initially, perfectly straight bars were modelled, although real geometries were always identified as imperfect. A significant advancement has been the introduction of finite element methods and geometrically and materially non-linear solutions [8] capable of analysing the load-bearing capacity of structures with initial imperfections; see, e.g., [9,10,11]. Generally, imperfections can be categorized into three main groups: geometrical imperfections, material imperfections, and structural imperfections [12].

Currently, the design of structural steel employs a systematic design approach known as the Direct Design Method [13], which explicitly accounts for material and geometric nonlinearities, residual stresses, and the presence of initial imperfections. The shape of these initial imperfections should be defined to take into account their influence on the designs’ load-carrying capacity while respecting the real-world measures from experiments; see, e.g., [14,15,16]. Typically, the first buckling mode shape is used to represent these initial imperfections, assuming it is the most critical [17,18]. However, in cases where the critical buckling loads of two distinct buckling modes coincide, the sensitivity to imperfections increases significantly [19,20]. In these cases, the buckling mode shapes may be more important than the magnitude of the critical force when modelling initial imperfections.

As most steel structures are unique, data on the initial imperfections from experiments are limited. One approach to modelling initial imperfections is using the superposition of the first several buckling modes and the probabilistic characteristics of their amplitudes [13,21]. Buckling modes have shapes of trigonometric functions with indeterminate amplitudes, which are a subject of research.

This article is based on the heuristic argument that if two distinct buckling modes coincide, the system tends to follow the buckling mode with lower energy and higher entropy. Entropy can provide new information about the stability behaviour of slender structures, both in the context of Euler buckling and geometrically nonlinear solutions with initial imperfections. Since the entropy of the buckling mode has not yet been studied, the concept of a surrogate model is proposed based on thermodynamics, where entropy naturally occurs. The entropy measure used in this study is defined by decomposing the strain energy of the buckling mode into entropy and virtual temperature.

Thus, a new concept of entropy and virtual temperature, based on the equivalence of strain and heat energy in the surrogate model, is introduced. The surrogate model is based on an unconventional connection between classical theories: entropy [22], heat energy [22], strain energy [23], and buckling [1]. Entropy is presented as a new indicator of stability and resistance to buckling. This study follows a method for modelling initial geometric imperfections using a linear combination of buckling modes scaled by entropy. The research progresses from a simple strut to more complex structures, such as steel plane frames.

## 2. Surrogate Entropy Model of Buckling

Structural mechanics uses the energies in the system, following the fundamental principles of Lagrangian mechanics, but does not consider entropy. The question is how to calculate entropy for a deformed structure. One of the approaches may involve a surrogate model based on thermodynamics and the isothermal process in ideal gases; see Figure 1.

Entropy can be viewed as a characteristic closely related to energy [24,25]. In thermodynamics, heat energy is generated as a result of changes in the entropy of a system [22]. In modern theories, this entropy can be described using a relationship based on the energy balance [26].
(1)F·Δx=T⋅ΔS.

Equation (1) relates mechanical work to thermodynamic work. The left-hand side of the equation is the mechanical work performed by force *F* on path Δ*x* as the body moves. The right-hand side of Equation (1) is the heat energy produced by the system at temperature *T* due to changes in the entropy of the system Δ*S*; see, e.g., [22].

Figure 1 illustrates the relationship between the strain energy of the lateral deformation of the strut and the heat energy of a surrogate model based on an ideal gas. The calculation of strain energy is grounded in linear elasticity theory, following Hooke’s law. The static load action of the critical force is taken into account. Heat energy is modelled using the work of pistons, where all pistons are identical, with an ideal gas, with constant mass and constant temperature, surrogating the original strut model. The pistons undergo deformation caused by buckling. The change in piston energy is equivalent to the change in energy due to bending in a buckled strut. Both energies depend on an indeterminate constant derived from Euler’s solution for buckling. With the exception of adopting buckling energy, the surrogate model, based on the isothermal process of the gas, does not exchange energy with its surroundings.

The entropic approach can be employed in novel applications, illustrated by modelling the initial imperfections through the utilization of scaled buckling modes.

## 3. Buckling of a Pin-Ended Strut

This section illustrates a pin-ended strut as a specific case for explaining the entropy problem. The strain energy is derived and decomposed into entropy and virtual temperature using a surrogate model. The section concludes with case studies of scaled buckling modes, showcasing the application of maximum entropy and minimum energy principles in the establishment of initial imperfections.

Consider a slender strut with pin-ended supports subjected to an axial compressive load *P*. The strut remains ideally straight until the load reaches its critical value *P_cr_* and buckling occurs [1,2]. The critical load places the strut in a state of unstable equilibrium, where, in addition to equilibrium on the straight strut, there also exists equilibrium on the deflected strut; see Figure 2.

For a strut under critical load, bending deformation can be described according to the Euler–Bernoulli beam theory, using the following differential equation:(2)$$\frac{dy^{2}\left(x\right)}{dx^{2}}+\frac{P}{E\cdot I}y=0,$$
where *y*(*x*) is the lateral deflection, *E* is Young’s modulus, and *I* is the second moment of the area. This equation is a linear, nonhomogeneous, differential equation of the second order with constant coefficients. Upon substituting *α*^2^ = *P*/(*E*·*I*), the particular solution of this differential equation can be expressed in the form
(3)$$y\left(x\right)=c_{1}\cdot \sin\left(\alpha \cdot x\right)+c_{2}\cdot \cos\left(\alpha \cdot x\right),\;\mathrm{for}\;x\in [0,\;L].$$

From the boundary conditions, it can be calculated that *c*_2_ = 0 for *y*(0) = 0 and *c*_1_·sin(*α*·*L*) = 0 for *y*(*L*) = 0, where *L* is the strut length and *c*_1_ is the indeterminate amplitude. By using *c*_1_ > 0, it can be obtained that sin(*α*·*L*) = 0, and thus *α* = *i*·π/*L*, where *i* represents a natural number of the buckling mode. The corresponding buckling modes can be derived from Equation (3) as eigenmodes, resulting in deformation patterns that are characterized by sine functions.
(4)$$y\left(x\right)=c_{1}\cdot \sin\left(i\frac{\pi \cdot x}{L}\right),\;\mathrm{for}\;x\in [0,\;L],\;i\;=\;1,\;2,\;\ldots ,$$
where *i* is the number of half-sine curvatures that occur lengthwise. Buckling modes can be described as the shapes the strut assumes during buckling, with the sine function playing a crucial role in their formulation; see Figure 3.

Using the previously defined equation *α*^2^ = *P*/(*E*·*I*), the critical loads are
(5)$$P_{cr,i}=i^{2}\cdot \pi^{2}\frac{E\cdot I}{L^{2}},\;\mathrm{for}\;i\;=\;1,\;2,\;\ldots ,$$

Equation (5) provides the standard critical load at which buckling occurs in a pin-ended strut. This provides insight into the behaviour of slender struts under axial compression, and valuable understanding of the design and analysis of structural systems.

In Euler buckling, the rank of the buckling modes is determined by the ranking of the smallest critical force to the largest. This is the most common approach for ranking buckling modes, with the first one considered the most dangerous [27].

### 3.1. Strain Energy

The estimation of entropy is based on the decomposition of the strain energy of a buckling mode into entropy and virtual temperature. In the first step, an analytical solution of the strain energy is derived.

The discretization of *y*(*x*) is performed at the centroids of finite elements. By introducing (*j* − 0.5)/*N* instead of *x*/*L*, Equation (4) can be rewritten in the discrete form
(6)$$f_{ij}=c_{1}\cdot \sin\left(i\frac{\pi \cdot \left(j-0.5\right)}{N}\right),\;\mathrm{for}\;j\;=\;1,\;2,\;\ldots \;N,$$
where *j* = 1, 2, … *N* and *N* is the number of beam elements. The elements are considered to have the same length.

In its differential form, the total internal potential energy (strain energy) of the *i*-th buckling mode can be obtained as
(7)$$\Delta \Pi_{i}=\frac{1}{2}\int_{0}^{L}E\cdot I{\left(\frac{dy^{2}\left(x\right)}{dx^{2}}\right)}^{2}dx=\frac{1}{2}\int_{0}^{L}E\cdot I{\left(c_{1}\cdot {\left(i\frac{\pi }{L}\right)}^{2}\cdot \sin\left(i\frac{\pi \cdot x}{L}\right)\right)}^{2}dx=i^{4}\cdot E\cdot I\cdot \frac{c_{1}^{2}\cdot \pi^{4}}{4\cdot L^{3}}.$$

The value of ΔΠ*_i_* represents the difference between the zero-strain energy of the unloaded strut and the strain energy of the buckled strut.

It can be noted that the square of the second derivative can be replaced by the product of the function value and the fourth derivative, *E*·*I*·(*y*″)^2^ = *y*·*E*·*I*·*y*⁗. In this expression, the term involving the fourth derivative, *E*·*I*·*y*⁗, represents a fictitious transverse load action.

The total potential energy is the sum of ΔΠ*_i_* and ΔΠ*_e_*. According to the principle of conservation of mechanical energy, the strain energy from the internal forces ΔΠ*_i_* (from bending moment) is equal to the potential energy of the external forces ΔΠ*_e_* (from load action *P_cr,i_*).
(8)$$\Delta \Pi_{e}=\frac{1}{2}\cdot P_{cr,i}\cdot \int_{0}^{L}{\left(\frac{dy\left(x\right)}{dx}\right)}^{2}dx=\frac{1}{2}\cdot i^{2}\cdot \pi^{2}\frac{E\cdot I}{L^{2}}\cdot \int_{0}^{L}{\left(c_{1}\cdot i\frac{\pi }{L}\cdot \cos\left(i\frac{\pi \cdot x}{L}\right)\right)}^{2}dx=i^{4}\cdot E\cdot I\cdot \frac{c_{1}^{2}\cdot \pi^{4}}{4\cdot L^{3}}.$$

The discrete form of Equation (7) is used to introduce discrete entropy. By introducing *L*/*N* instead of *dx* and *j*/*N* instead of *x*/*L*, Equation (7) can be rewritten in its discrete form as
(9)$$\Delta \Pi_{i}=\sum_{j=1}^{N}\Delta \Pi_{ij}=\frac{1}{2}\sum_{j=1}^{N}E\cdot I\cdot {\left(c_{1}\cdot {\left(i\frac{\pi }{L}\right)}^{2}\cdot \sin\left(i\frac{\pi \cdot \left(j-0.5\right)}{N}\right)\right)}^{2}\frac{L}{N},\;\mathrm{for}\;N\;>\;i,$$
when *N* > *i* and *i* is the index of the buckling mode. The discrete form is suitable for use in the finite element method. One member of the series ΔΠ*_ij_* is the strain energy of a single element of the strut. The square of the sine function in Equation (9) converges to the value of *N*/2.
(10)$$\sum_{j=1}^{N}{\left(\sin\left(i\frac{\pi \cdot \left(j-0.5\right)}{N}\right)\right)}^{2}=\frac{N}{2},\;\mathrm{for}\;N\;>\;i.$$

Let each buckling mode have its own scale factor using amplitude *c_i_*. By substituting Equation (10) into Equation (9), the sum of the discrete strain energy is equal to the value calculated from Equation (7).
(11)$$\Delta \Pi_{i}=i^{4}\cdot E\cdot I\cdot \frac{c_{i}^{2}\cdot \pi^{4}}{4\cdot L^{3}},$$
where *i* is the buckling mode number, *E* is the elastic Young’s modulus, *I* is the second moment of the area, *L* is the length of the specimen, and *c_i_* is the amplitude of the half-sine curvature that occurs lengthwise.

The strain energy is a function of the indeterminate amplitude *c_i_*, as seen, for example, in Equation (11). However, if it is possible to link energy with entropy, ranking based on the strain energy can be justified.

### 3.2. Entropy in the Surrogate Model

The strain energy of the deformed structure can be decomposed into temperature and entropy using a surrogate model. For this purpose, the strain energy is transformed into gas energy. In investigating the characteristics of a surrogate model, the terms entropy Δ*S_i_* and virtual temperature *T_i_* are used. Following Equation (1), it can be written as
(12)$$\Delta \Pi_{i}=T_{i}\cdot \Delta S_{i},$$
where ΔΠ*_i_* is the strain energy of the buckled strut from the previous section, and the right-hand side is the heat energy associated with the gas surrogate model from the following section.

When the virtual temperature *T_i_* is constant, the change in entropy Δ*S* can be expressed as the work of an ideal gas with constant mass *m* during an isothermal process. Under constant *T_i_*, the internal energy of the gas remains unchanged. Thus, the heat absorbed during pressure change equals the work done by the gas; see, e.g., [28]. The entire system is isolated, meaning there is no exchange of particles or energy with the surrounding environment, i.e., the number of particles and energy remain constant.

The change in entropy of an isothermal process for gases, where pressure varies as a function of volume, can be expressed as
(13)$$\Delta S=\frac{m\cdot R}{M_{m}}\ln\frac{\rho_{1}}{\rho_{2}}=\frac{m\cdot R}{M_{m}}\ln\frac{V_{2}}{V_{1}},$$
where *ρ*_1_ is the initial pressure, *ρ*_2_ is the final pressure, *V*_1_ is the initial volume, *V*_2_ is the final volume, *M_m_* is the molar mass of the gas, and *R* = 8.314 Jmol^−1^K^−1^ is the molar gas constant; see, e.g., [29].

The change in entropy in the surrogate model can be described using the change in gas volume. The buckling causes gas compression (volume reduction) on the deflection side and expansion (volume increase) on the opposite side of the strut; see Figure 4. The pistons are located at the centroids of the finite elements in the direction of deformation, and the mesh of the finite elements is uniform.

The change in entropy Δ*S_ij_* can be written on the basis of Equation (13). The gas loading of the *j*-th element (compression and expansion) has a mass of *m*/*N*, where *N* is the number of beam elements. The mass of the gas in one piston is 0.5·*m*/*N*. For the *j*-th element, the equation for the change in entropy on one side of the strut (compression or expansion) can be obtained via
(14)$$\Delta {\bar{S}}_{ij}=\frac{m\cdot R}{2\cdot N\cdot M_{m}}\ln\frac{V_{ij}}{V_{1}}=\frac{m\cdot R}{2\cdot N\cdot M_{m}}\ln\frac{A\cdot h_{ij}}{A\cdot h}=\frac{m\cdot R}{2\cdot N\cdot M_{m}}\ln\frac{h_{ij}}{h},$$
where *V_ij_* is the final gas volume and *h_ij_* is the final piston height. The volume *V* is expressed as the product of the area *A* and the height *h*. The entropy Δ*S_ij_*, depending on two indices, is similar to the volumes of the pistons *V_ij_*. The change in volume can be expressed as the change in the height *h* of the piston in the gas vessel, where *h* is a constant.

Figure 4 presents the gas analogy with virtual forces that straighten the elastic strut released by the axial force. The change in the entropy of the gas on both sides of the beam element can be expressed as the difference between two entropies caused by expansion (*h* + Δ*h_ij_*)/*h* and compression (*h* − Δ*h_ij_*)/*h*.
(15)$$\Delta S_{ij}=\Delta S_{ij}^{E}-\Delta S_{ij}^{C}=\frac{m\cdot R}{2\cdot N\cdot M_{m}}\left(\ln\frac{h+\Delta h_{ij}}{h}-\ln\frac{h-\Delta h_{ij}}{h}\right)\approx \frac{m\cdot R}{N\cdot M_{m}}\cdot \frac{\Delta h_{ij}}{h},$$
where displacement Δ*h_ij_* represents a small change in the piston height. The characteristic Δ*h_ij_*/*h* can be interpreted as an analogy to the strain *ε* in Hooke’s laws. The small displacements are a standard condition in the Euler–Bernoulli beam theory in structural mechanics. The magnitude of Δ*S_ij_* can be expressed as
(16)$$\Delta S_{ij}\approx \frac{m\cdot R}{h\cdot M_{m}}\cdot \frac{1}{N}\cdot \left|\Delta h_{ij}\right|=k_{1}\cdot \frac{1}{N}\cdot \left|\Delta h_{ij}\right|,\;\mathrm{where}\;\Delta h_{ij}\;<<\;h.$$

This equation expresses that there is an entropy change in the direction of deformation Δ*h_ij_*. The constant *k*_1_ replaces the characteristics of the gas in the pistons, and in the context of the buckling of the strut, it represents lateral deformation stiffness. The change in entropy is linearly dependent on displacement.

### 3.3. Entropy and Virtual Temperature

During the *i*-th buckling mode, the virtual temperature *T_i_* is constant in the isothermal process, and the strain energy can be summed over all beam elements, where *T_i_* is constant across all elements. In Equation (12), entropy and strain energy are additive quantities, so their values can be obtained by the summation of all elements. After introducing Δ*h_ij_* = *f_ij_*, the summation of all *j*-th elements leads to
(17)$$\underset{\Delta \Pi_{i}}{\underset{︸}{i^{4}\cdot E\cdot I\cdot \frac{c_{i}^{2}\cdot \pi^{4}}{2\cdot L^{3}}\cdot \frac{1}{N}\sum_{j=1}^{N}{\left[\sin\left(i\frac{\pi \cdot \left(j-0.5\right)}{N}\right)\right]}^{2}}}=T_{i}\cdot \underset{\Delta S_{i}}{\underset{︸}{k_{1}\cdot \frac{1}{N}\cdot \sum_{j=1}^{N}\left|c_{i}\cdot \sin\left(i\frac{\pi \cdot \left(j-0.5\right)}{N}\right)\right|}},$$
where *c_i_* > 0 is a scale factor of the *i*-th buckling mode. Equation (17) creates a connection between the products of the second derivative (bending moment) on the left-hand side and the product of deformation on the right-hand side of the equation. The sum of the absolute value of the sine function in Equation (17) converges to the value of 2·*N*/*π*.
(18)$$\underset{N\to \infty }{\lim}\sum_{j=1}^{N}\left|c_{i}\cdot \sin\left(i\frac{\pi \cdot \left(j-0.5\right)}{N}\right)\right|=2\cdot N\cdot \frac{c_{i}}{\pi }.$$

For *i* < *N* < ∞, Equation (18) takes the form of an approximate relationship. By substituting Equations (10) and (18) into Equation (17), Equation (17) can be simplified to the form of Equation (19).
(19)$$\underset{\Delta \Pi_{i}}{\underset{︸}{i^{4}\cdot E\cdot I\cdot \frac{c_{i}^{2}\cdot \pi^{4}}{4\cdot L^{3}}}}=T_{i}\cdot \underset{\Delta S_{i}}{\underset{︸}{2\cdot k_{1}\cdot \frac{c_{i}}{\pi }}},$$
where the left-hand side is the change in strain energy ΔΠ*_i_* and the right-hand side *T_i_*·Δ*S_i_* takes into account the change in entropy. By separating the entropic term from Equation (20), an expression for entropy can be written.
(20)$$\Delta S_{i}=2\cdot k_{1}\cdot \frac{c_{i}}{\pi },$$
where *m* is the mass of the gas in all pistons and *m*, *R*, *M_m_*, *h* are constants. Equation (20) expresses the entropy of the ideal gas in the surrogate model. Due to the specific shape of the sine functions of the buckling modes, the entropy of the ideal gas is a function of only the amplitude *c_i_* and constant *k*_1_.

The virtual temperature *T_i_* of the gas calculated from the surrogate model of the pin-ended strut can be written using Equation (19), as
(21)$$T_{i}=i^{4}\cdot c_{i}\cdot E\cdot I\cdot \frac{\pi^{5}}{8\cdot k_{1}\cdot L^{3}}.$$

The decomposition of ΔΠ*_i_* into *T_i_* and Δ*S_i_* introduced both *T_i_* and Δ*S_i_* as dependent on *c_i_*, but only the virtual temperature *T_i_* is a fourth power function of the index *i*.

Equation (19) can be written using *P_cr,i,_*
(22)$$\underset{\Delta \Pi_{i}}{\underset{︸}{\overset{P_{cr,1}}{\overset{︷}{\pi^{2}\frac{E\cdot I}{L^{2}}}}\cdot i^{4}\cdot \frac{c_{i}^{2}\cdot \pi^{2}}{4\cdot L}}}=\underset{T_{i}}{\underset{︸}{\overset{P_{cr,1}}{\overset{︷}{\pi^{2}\frac{E\cdot I}{L^{2}}}}\cdot i^{4}\cdot \frac{c_{i}\cdot \pi^{3}}{8\cdot k_{1}\cdot L}}}\cdot \underset{\Delta S_{i}}{\underset{︸}{2\cdot k_{1}\cdot \frac{c_{i}}{\pi }}}.$$

Equation (22) introduced the decomposition of the strain energy ΔΠ*_i_* into terms of virtual temperature *T_i_* and entropy Δ*S_i_*. The utilization of Equation (22) can be presented in two fundamental cases.

In thermodynamics, any equilibrium state can be characterized either as a state of maximum entropy for a given energy or as a state of minimum energy for a given entropy [30]. In structural mechanics, the strain energy and the principle of minimum total potential energy exist in many applications [31,32,33,34,35,36,37,38,39,40], but the principle of maximum entropy is not commonly considered. Equation (22) establishes a link between structural mechanics and thermodynamics. In the surrogate model, the principle of maximum entropy can be applied using the right-hand side of Equation (22).

The case of constant energy, ΔΠ*_i_* = constant. If *c_i_* = *c*_1_/*i*^2^ is introduced, then the strain energy remains constant across all buckling modes, and entropy decreases with the square of index *i*. The first buckling mode, which has the highest entropy at a constant energy, is realized first.

In the case of constant entropy, Δ*S_i_* = constant: if *c_i_* = *c*_1_ is introduced as a constant for all buckling modes, then the strain energy increases with the fourth power of the index *i*, and entropy remains constant across all buckling modes. The first buckling mode, which has the lowest strain energy at constant entropy, is realized first.

### 3.4. The Case Study

This case study considers a pin-ended IPE240 steel member with the length of *L* = 3 m. The member has a Young’s modulus of *E* = 210 GPa, and its second moment of area is *I* = 2.83 · 10^−6^ m^4^. Euler’s critical load is *P_cr,_*_1_ = 651.7 kN. The results in Table 1 are obtained with the assumption of *c_i_* = 1 m and *k*_1_ = 1000 Jm^−1^K^−1^. Table 1 presents a scenario of constant entropy, as entropy does not depend on the buckling mode index. 

In Equation (22), considering the decomposition of strain energy into entropy, virtual temperature is the measure by which energy is evaluated in terms of entropy.

The criterion that the equilibrium state can be characterized as a state of minimum energy for given entropy can be applied. The minimal strain energy occurs for the first buckling mode, which occurs first; see Table 1. Subsequent buckling modes are ranked in ascending order, just as if according to critical forces. It holds that ΔΠ*_i_* = *i*^4^·ΔΠ_1_.

The same conclusion can be obtained using the criterion that the equilibrium state can be characterized as a state of maximum entropy for a given energy. By using Equation (22), the amplitude (scale factor) needs to be set as decreasing across buckling modes, *c_i_* = *c*_1_/*i*^2^, in order to keep the strain energy constant across all modes. Using this new scale, the entropy of the buckling modes is computed; see the last column in Table 2.

In Table 2, the criterion of maximum entropy for a given energy is applied. The first buckling mode, which occurs first, corresponds to the maximum entropy. Subsequent buckling modes are ranked in descending order, inversely to their ranking by critical forces.

The scale factor *c_i_* in Table 2 decreases approximately as intensively as the mean values of the scale factors of the first three buckling modes for I-sections in [21]. The article [21] presents data in its Table 1 from measurements of the initial out-of-straightness of nine IPE 160 columns, performed at the Polytechnic University of Milan and published by the ECCS 8.1 committee [41]. In addition, the article [21] uses the results from the measurements of 428 samples [42]. The scale factors *c_i_* in Table 2, obtained from the analysis of strain energy and entropy, are practically the same (differences are minimal) as the scale factors from those experiments [21,41].

The case in Table 2 is a realistic reflection of the buckling problem, and the scale factor *c_i_* represents the imperfection measure. Thus, the initial imperfection can be introduced as a linear combination of the scaled buckling modes using their amplitudes *c_i_* = *c*_1_/*i*^2^, where *I* is the index of the buckling mode and *c*_1_ is the amplitude of the first buckling mode. The amplitudes *c_i_* also express the (virtual) entropy for a given strain energy.

The ranking of buckling modes by strain energy in Table 1 and by entropy in Table 2 is the same. In the case study, both the criteria of minimum energy and maximum entropy lead to the same ranking of the buckling modes.

## 4. Buckling of a Cantilever

It can be assumed that the entropy in Equation (16) is applicable to other patterns of the buckling modes of other compression columns with different boundary conditions. The entropy of the *i*-th buckling mode can be obtained by summing the terms from Equation (16).
(23)$$\Delta S_{i}=k_{1}\cdot \frac{1}{N}\cdot \sum_{j=1}^{N}\left|\Delta h_{ij}\right|,$$
where *k*_1_ is constant and *h_ij_* is a deflection from the eigenvalue vector of the *i*-th buckling mode. Each eigenvector is dimensionless, with an indeterminate scale factor *c_i_*. For example, the deformation of the *i*-th buckling mode of a cantilever can be expressed by the function
(24)ycx=ci⋅1−cos2i−1π⋅x2⋅L, for x∈[0, L], i = 1, 2, …,
where *c_i_* > 0 and the prefix *c* in *^c^y*(*x*) stands for cantilever. By substituting *j*/*N* for *x*/*L*, the continuous function can be written in its discrete form as
(25)fcij=ci⋅1−cos2i−1π⋅j−0.52⋅N, for j = 1, 2, … N.

For Δ*h_ij_* = *^c^f_ij_*, the entropy of the *i*-th buckling mode of the cantilever can be written as
(26)ΔcSi=k1⋅1N⋅∑j=1Nci⋅1−cos2i−1π⋅j−0.52⋅N.

In contrast to a pin-ended strut, the entropy of cantilever buckling modes is not constant but slightly dependent on the *i*-th index, especially for the first buckling mode.
(27)ΔcSi=k1⋅ci⋅1+cosπ⋅iπ⋅i−0.5,
where [·] is not constant across buckling modes. However, the change in entropy is small compared to the change in strain energy. The change in strain energy ΔΠ*_i_* of the *i*-th buckling mode of the cantilever can be derived analogously to the solution in Section 3.
(28)ΔcΠi=12⋅∫0LE⋅Idy2xdx22dx=E⋅I⋅π4⋅ci264⋅L3⋅2⋅i−14.

By substituting Equations (27) and (28) into Equation (12), the virtual temperature *^c^T_i_* can be written as
(29)Tci=ΔcΠiΔcSi=2⋅i−14⋅ci⋅E⋅I⋅π464⋅k1⋅L3⋅π⋅i−0.5π⋅i−0.5+cosπ⋅i.

If *c_i_* = *c*_2_ is introduced as a constant for all buckling modes, the strain energy greatly increases with the fourth power of index *i*. However, the entropy computed from the surrogate model changes slightly across the buckling modes.

Similarly to the pin-ended strut, the strain energy increases significantly with the fourth power of the index *i*. In comparison to the pin-ended strut, the entropy calculated from the surrogate model is slightly dependent on the index *i*, as shown in Equation (27) compared to Equation (20). This example implies that the entropy of other members with different supports may also exhibit a slight dependence on the index *i*. The virtual temperature is proportional to the fourth power of the index *i*, as indicated by Equations (29) and (21).

In summary, it can be concluded that the dependencies of the pin-ended strut and cantilever are very similar. Generalized to other structures, entropy may show a slight dependence on the index *i*, while the virtual temperature exhibits a strong dependence on the fourth power of the index *i*. Both entropy and virtual temperature are linearly dependent on the amplitude *c*, as observed in both the pin-ended strut and cantilever cases.

## 5. Buckling of Steel Plane Frames

The design criteria for steel structures are built on the principles of the natural sciences and empirical experience [2]. The limit states of structures are traditionally studied using the finite element method, which allows for modelling structures with general geometry, boundary conditions, and loading. Engineers combine theoretical knowledge with practical expertise and innovations to design safe and reliable structures.

In the previous sections, strain energy was decomposed into its components of virtual temperature and entropy. This chapter consists of a case study that verifies the heuristic argument that a similar decomposition can be carried out for steel plane frames, albeit with certain limitations. In the initial step, the strain energy and its dependence on the index *i* of the buckling mode are studied.

The change in the strain energy for the *i*-th elastic buckling mode of the steel plane frame can be calculated using a second-order theory-based finite element model (FEM); see, e.g., [20].
(30)$$\Delta \Pi_{i}=\sum_{k=1}^{K}\left[\frac{1}{2}\cdot \int_{0}^{L_{k}}E\cdot I_{k}{\left(\kappa_{i}\cdot \frac{dy_{k}^{2}\left(x\right)}{dx^{2}}\right)}^{2}dx\right],$$
where *L_k_* is the length of the *k*-th member, *I_k_* is the second moment of the area of the *k*-th member, and *y_k_*(*x*) represents the deformation of the *i*-th elastic buckling mode of the *k*-th member. The input condition for addressing Equation (30) involves incorporating eigenmodes normalized to the sum of deformations equal to one.

A significant gap exists regarding the calculation of entropy change in buckled frames. Without a consideration of entropy, strain energy is analysed using an undetermined scale, denoted as *κ_i_* in Equation (30). It can be assumed that, for a given strain energy, the entropy will decrease analogously to the results in Table 2. The scale factor *κ_i_* can be calibrated in such a way that the strain energy of each buckling mode is equal to the strain energy of the first mode. The most critical mode should have the highest entropy corresponding to the lowest critical force, but this question has not been examined so far.

The methodology for ranking the buckling modes of steel plane frames can be illustrated in a case study. The columns and cross-beam of the frame are made of hot-rolled IPE240 members. The structural material is steel grade S235, with a Young’s modulus of elasticity *E* = 210 GPa, and the second moment of the area for the IPE profile is *I_y_* = 3890 cm^4^. The geometry of the frame is shown in Figure 5.

The frame is modelled using the finite element method, with the stiffness matrices and geometric matrix of the beam finite element published in [20]. The frame is meshed using 37 nodes and 36 beam finite elements. Each element has nine internal points where deformations are utilized. In total, there are 37 + 36 · 9 = 361 deformation points, where each point has one horizontal and one vertical deformation. The total deformation at each point, which has a general direction, is calculated using the Pythagorean theorem. Rotations are not considered. The vector of all 361 total deformations is normalized so that the sum of all total deformations equals one. The critical forces and corresponding buckling modes are computed using a second-order theory-based FEM [20]; see Figure 6.

The first six eigenvalues (critical forces) and eigenvector deformations (buckling modes) are computed using the step-by-step loading method; see Figure 7.

The procedure was practically carried out as follows: The critical forces *P_cr,i_* and the corresponding buckling modes (eigenvectors) were computed. Using *κ_i_* = 1 and the normalized scale of total deformations, the strain energies of the buckling modes were computed; see the third column in Table 3. The strain energy from the third column approximately correlates with the strain energy calculated using the approximation formula *i*^4^ ΔΠ_1_, where the formula *i*^4^ ΔΠ_1_ was explicitly derived for pin-ended struts.

Equation (30) uses the scale factor *κ_i_*, which influences ΔΠ*_i_*. If the value of *κ_i_* = 1, then ΔΠ*_i_* is solely computed from the normalized total deformations; see Table 3.

The last column in Table 4 provides the approximation of entropy ~ *κ_i_*, which sets the strain energy of the *i*-th buckling mode as equal to the strain energy of the first buckling mode. The value of *κ_i_* is calculated according to Equation (31).
(31)$$\kappa_{i}=\sqrt{\frac{\Delta \Pi_{1}}{\Delta \Pi_{i}}}$$
where the strain energies under the square root are calculated solely from the normalized total deformations.

The strain energy can be roughly approximated as ΔΠ*_i_* ≈ *i*^4^ ΔΠ_1_. Similarly to the strain energy relationship between the frame and the pin-ended strut, a similar relationship for entropy can be assumed. Although the link is not direct, it can be expected that the change in entropy can be found to be linear in *κ_i_*, albeit with the potential influence of additional factors. Although the equation for computing the frame’s entropy has not been derived, a strong correlation between entropy (for a given energy) and the results in the last column of Table 4 can be expected. The scale factor *κ_i_* in the penultimate column can be heuristically considered as the entropy estimate; see Table 4.

The perfect independence of entropy from the index *i* exists only for pin-ended struts; see Equation (20). The distinction between a frame and a strut lies in their structural composition. A frame consists of distinct and differently loaded structural members, with each absorbing a different part of the strain energy. For instance, a frame may include an unloaded cantilever that can buckle without strain. The significant presence of such structural elements represents a limitation of this method. In such cases, the frame’s entropy is slightly dependent on the index *i*, because entropy is associated with deformation due to buckling. In this context, entropy computed from the normalized eigenvector can be understood as an estimation based on the assumption that the behaviour of the frame is similar to pin-ended struts.

If there is an estimate of entropy, the criterion that the equilibrium state can be characterized as a state of maximum entropy for a given energy can be applied. The maximum entropy occurs for the first buckling mode, which occurs first; see the last column in Table 4. The same conclusion can be obtained using the criterion of the minimum energy for the given entropy; see the third column of Table 3. It can be noted that this conclusion applies only to the case studies presented in this article, and it is not generally proven for every frame that the ranking of buckling modes by entropy must be the same as their ranking by critical force. For instance, if the first and second critical buckling loads coincide, it may be useful to explore alternative approaches for the ranking of buckling modes [19].

This section addressed Euler stability, which investigates the buckling of ideal frames without initial imperfections. However, real structures have initial geometric imperfections due to their manufacturing and assembly. The shape and magnitude of these imperfections are still a matter of discussion, which will be the subject of the next section.

## 6. Initial Geometrical Imperfections

This chapter is based on the idea that initial geometric imperfections, simulating the imperfections characteristic of a physical structure, can be modelled as a linear combination of scaled buckling modes [43]. The load-carrying capacity of the structure is then computed using a geometric nonlinearity solution. The challenge lies in estimating the magnitudes of the buckling modes that should probabilistically capture the tendency of the frame to exhibit individual buckling shapes [13,21,43]. This article introduces entropy as a measure of this statistical tendency.

The classical method uses the scaling of the lowest eigenmode, see, e.g., [44,45]. The concept behind this theory is that the most critical imperfect geometry is the closest to the final collapse configuration, since it requires the least deformation energy to go from an unloaded state to its final collapse situation [46].

The initial imperfections of frame structures can be based on scaled buckling modes. Such fittings of randomly generated shapes of imperfection have been introduced, e.g., in [43]. This method is based on the scale factors of cold-formed steel members calculated on the basis of experimental measurements [47]. The use of the energy measure of eigenmodes to introduce initial imperfections has been presented for cold-formed steel columns [48] and axially compressed cylindrical shells [49]. In the case of shells, the energy measure of the geometric imperfections was defined by the square root of the strain energy [48].

The advanced introduction of the initial frame imperfections using entropy as a supporting energy factor has not yet been presented. Entropy, as a complement to strain energy, plays an important role in structural stability analysis by justifying the scaling of the buckling modes based on strain energy. The case study presented here demonstrated the rational scales of the buckling modes, with the dominant position of the first buckling mode. Ranking by strain energy is possible when the entropy remains constant across all buckling modes, and, analogously, ranking by entropy is achievable only under constant strain energy.

The scales of the buckling modes exhibit a decreasing trend if the same strain energy is considered in each buckling mode; see Table 3. Such scaled buckling modes can be applied in the modelling of initial geometric imperfections. Examples of initial geometric imperfection shapes using combinations of the scaled buckling modes are shown in Figure 8.

The signs of the scale factors of the buckling modes create 2^6^ = 64 combinations. The first buckling mode is dominant and the other modes only slightly change its shape; see Figure 8. However, small changes in the initial geometry of the columns can significantly influence the load-carrying capacity of the steel plane frame. The initial imperfection of the cross-beam is not important because it is subjected to a bending moment due to the lateral deformations of the columns. Compared to other geometrical and material imperfections, the most significant sensitivity arises when the non-dimensional slenderness of the columns is approximately equal to one, as demonstrated via global sensitivity analysis [18].

The basic assumption is that enough modes must be applied to ensure that the deformations of critical elements are induced by imperfections [21]. However, the scale factors of higher-order modes decrease rapidly and their contribution can be neglected from a certain order. It can be noted that common engineering practice often uses only the first buckling mode [21]. To estimate the limit state, the initial imperfection should be close to the critical mode, which replaces the shape of the frame in the limit state with failure due to buckling. This approach is consistent with the EUROCODE 3 design standard [27].

In the stochastic approach, the initial imperfection can be thought of as an approximation of the real imperfection from measurements on a large number of frames. Scaled buckling modes can be used for the random simulation of the shapes of the initial imperfections. One possibility is to consider the contributions of buckling modes with a mean value of zero (a perfect strain frame) and random variability based on the scale factor. A similar approach has been applied in probabilistic reliability analyses [18].

## 7. Conclusions

This article investigated the duality of minimum energy and maximum entropy in the context of buckling. The entropy analysis of structures supports energy-based computational results and provides a rationale for ranking buckling modes based on strain energy. An explicit solution was derived for the relationship between the strain energy and change in entropy within the context of the pin-ended strut. Strain energy is decomposed into virtual temperature and entropy using a surrogate model. The change in entropy was found to be linear in displacement. The principle of this decomposition is generalized for steel plane frames, where it is applied as an approximate relationship.

A contribution to engineering practice is the introduction of an entropy-based analysis of buckling modes. The introduction of entropy extends the applicability of the criterion of minimum energy in structural mechanics. Buckling modes can be ranked according to their strain energy for the given entropy. Alternatively, buckling modes can be ranked according to their entropy for the given strain energy. The case studies have shown that the ranking aligns with the traditional method based on critical forces, but this conclusion cannot be generalized for all types of frames and can be studied further.

Another application of scaled buckling modes was found in the analysis of the initial geometric imperfections of steel columns. The scaled buckling modes, based on a constant energy, approximately coincide with the experimentally determined initial imperfections of steel IPE160 columns [21,41]. The scale factors can be interpreted as an estimate of entropy. This observation enables the inverse modelling of initial imperfections using scaled buckling modes, where each mode has the same strain energy.

In the case of frame structures, the scale factors of the buckling modes demonstrate a decreasing trend when the strain energy is constant in each mode. These scale factors can heuristically be interpreted as an estimate of entropy. Scaled buckling modes are valuable when dealing with the initial geometric imperfections formed by their linear combination. Each buckling mode is assigned the same strain energy through a change in scale. The initial imperfection can be considered a linear combination of these scaled buckling modes, similar to the case of columns.

## Figures and Tables

**Figure 1 entropy-25-01630-f001:**
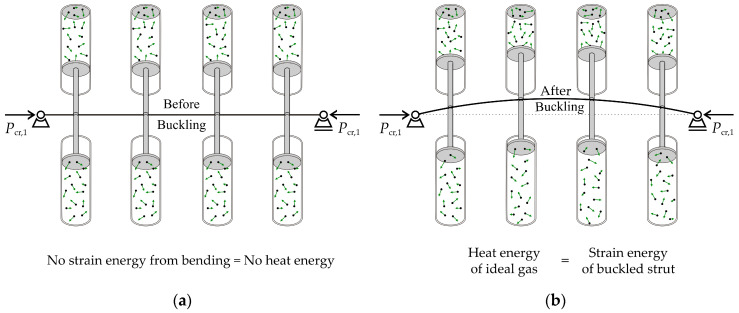
Surrogate model with ideal gas. (**a**) State before buckling. (**b**) State after buckling; a pair of pistons has a resultant non-zero force.

**Figure 2 entropy-25-01630-f002:**
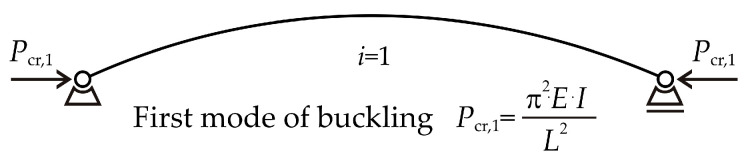
The flexural buckling of the pin-ended strut.

**Figure 3 entropy-25-01630-f003:**
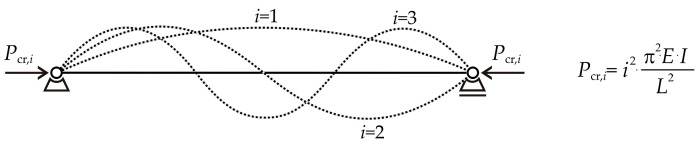
The buckling modes of the pin-ended strut.

**Figure 4 entropy-25-01630-f004:**
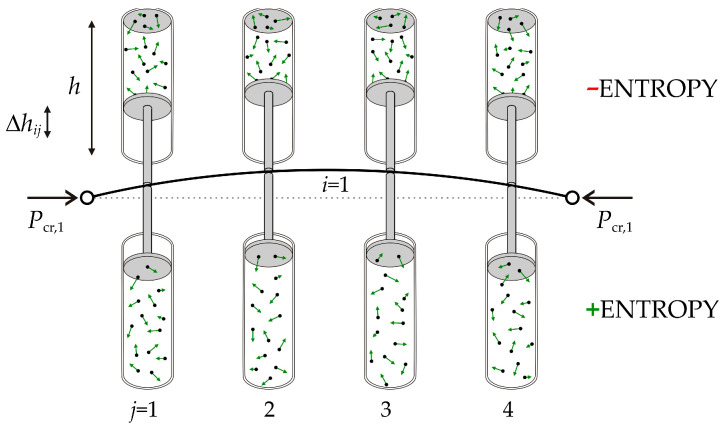
Surrogate model for the transformation of the buckling mode into entropy.

**Figure 5 entropy-25-01630-f005:**
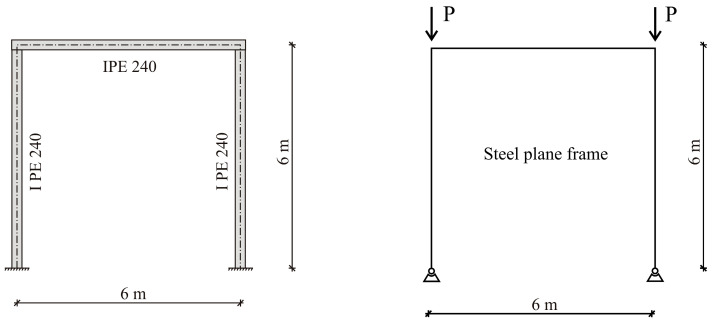
The geometry of the steel plane frame.

**Figure 6 entropy-25-01630-f006:**
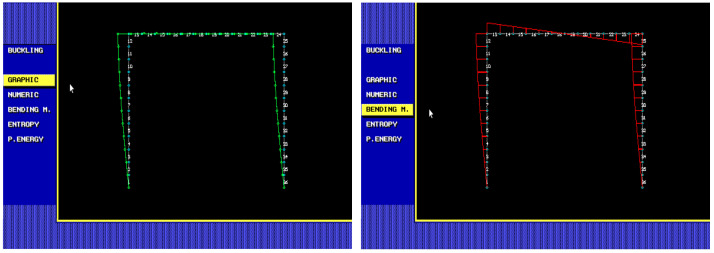
Second-order theory-based FEM: first buckling mode and bending moment of a steel plane frame.

**Figure 7 entropy-25-01630-f007:**
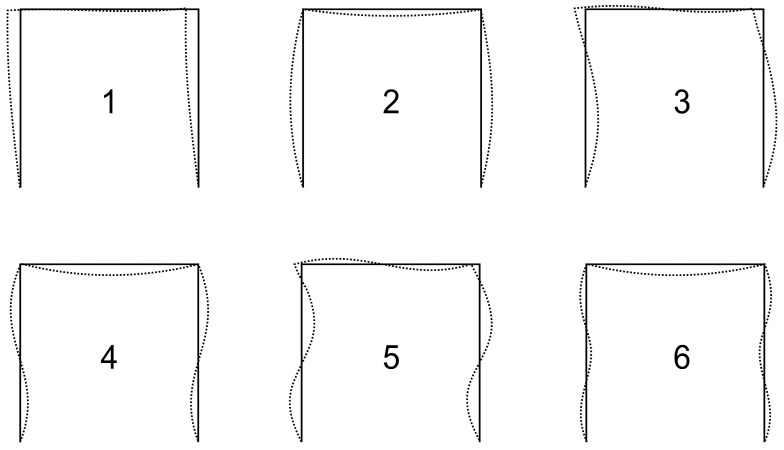
First six buckling modes of a steel plane frame.

**Figure 8 entropy-25-01630-f008:**
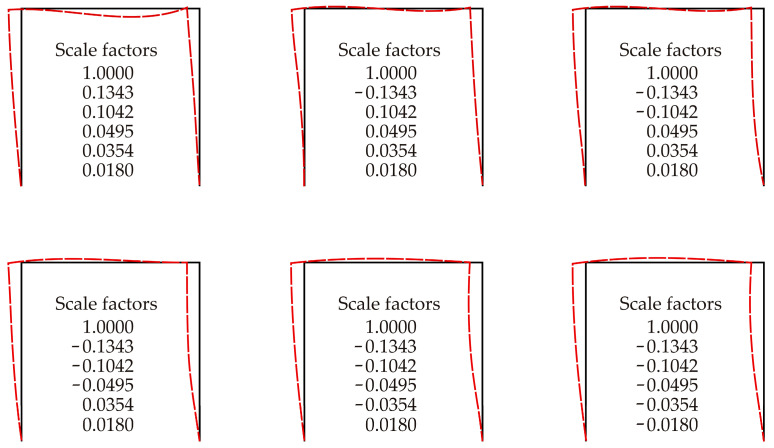
Initial imperfections from scaled buckling modes, first six combinations.

**Table 1 entropy-25-01630-t001:** Ranking of buckling modes according to strain energy, assuming constant entropy.

Buckling Mode Index	Critical Force *P_cr,i_* [kN]	*c_i_*	Strain Energy ΔΠ*_i_* [MJ]	Virtual Temperature *T_i_* [K]	Entropy Δ*S_i_* [J·K^−1^]
1	651.7	1	0.536	842.0	636.62
2	2606.9	1	8.576	13,472.7	636.62
3	5865.5	1	43.418	68,200.3	636.62
4	10,427.6	1	137.221	215,546.7	636.62
5	16,293.1	1	335.013	526,237.0	636.62
6	23,462.0	1	694.683	1,091,205.0	636.62

**Table 2 entropy-25-01630-t002:** Ranking of buckling modes according to entropy, assuming constant strain energy.

Buckling Mode Index	Critical Force *P_cr,i_* [kN]	*c_i_*	Strain Energy ΔΠ*_i_* [MJ]	Virtual Temperature *T_i_* [K]	Entropy Δ*S_i_* [J·K^−1^]
1	651.7	1	0.536	1.0000	636.62
2	2606.9	0.25	0.536	15.3229	159.15
3	5865.5	0.111	0.536	25.0961	70.74
4	10,427.6	0.063	0.536	58.2375	39.79
5	16,293.1	0.04	0.536	72.6713	25.46
6	23,462.0	0.028	0.536	130.7333	17.68

**Table 3 entropy-25-01630-t003:** Ranking of the buckling modes according to the strain energy for a given entropy.

Buckling Mode Index	Critical Force *P_cr,i_* [kN]	Strain Energy ΔΠ*_i_* [J] (from Normalized Eigenmodes)	Scale Factor *κ_i_* (~Entropy)	Approximation of Strain Energy ≈*i*^4^ ΔΠ_1_ [J]
1	412.5	1.1056	1.0000	1.1056
2	2925.8	61.3416	1.0000	17.6905
3	3830.9	101.8158	1.0000	89.5581
4	9784.0	451.7487	1.0000	283.0477
5	11,085.0	880.4734	1.0000	691.0345
6	21,023.4	1406.9822	1.0000	1432.9292

**Table 4 entropy-25-01630-t004:** Ranking of the buckling modes according to the entropy for a given strain energy.

Buckling Mode Index	Critical Force *P_cr,i_* [kN]	Given Strain Energy ΔΠ*_i_* [J] (by Scaled Eigenmodes)	Scale Factor *κ_i_* (for ΔΠ*_i_* = ΔΠ_1_)	Entropy ~ *κ_i_*
1	412.5	1.1056	1.0000	1.0000
2	2925.8	1.1056	0.1343	0.1343
3	3830.9	1.1056	0.1042	0.1042
4	9784.0	1.1056	0.0495	0.0495
5	11,085.0	1.1056	0.0354	0.0354
6	21,023.4	1.1056	0.0180	0.0180

## Data Availability

Data are contained within the article.

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
