# Peer review of "Strain Energy and Entropy Based Scaling of Buckling Modes"

_entropy, 2023, doi:10.3390/e25121630_

Round 1

Reviewer 1 Report

Comments and Suggestions for Authors

In the paper a new utilization of entropy in the context of buckling is presented. The novel concept connecting strain energy and entropy for a pin-ended strut is derived.

 The article is highly up-to-date and valuable for both theory and practice. I recommend publishing in the form presented.

​

Author Response

Thank you for the time you spent considering my article for publication in Entropy.

Reviewer 2 Report

Comments and Suggestions for Authors

The following comments would help improve the presentation quality:

1) The presentation structure requires substantial improvements:

  • Section 2 presents bibliography information irrelevant to the considered problem. Sections 3.2 and 3.3 are also uninformative and located in the wrong place. Please also avoid unclear statements, e.g., “If it is possible to decompose strain energy using entropy, then ranking buckling modes based on energy will be possible. By assigning the same entropy to all buckling modes, buckling modes can be ranked by strain energy (from the smallest to the largest).” What does it mean? How does it work? The application example can clarify the issue, but this text is irrelevant in the present place. So, the Reviewer recommends deleting Section 2 for the presentation simplicity. Instead of this deleted text, the Author should introduce the surrogate entropy model. The Reviewer also disagrees with the indirect relationship between the surrogate gas model and deformation analysis mentioned in the text (e.g., Line 278). This Expert sees the gas analogy (Figure 4) with virtual forces, which straighten the elastic strut released of the axial force. The description in Lines 251–256 is also essential and should precede the theoretical model. Therefore, the updated Section 2 must clarify the modeling analogy and formulate the analysis assumptions. On the other hand, the application examples must explain the buckling mode ranking procedure. Moreover, this ranking is unimportant; it only makes the imperfection analysis possible. Please put the correct accents in the presentation.
  • The Title determines the “initial imperfections” as the primary issue. Unfortunately, the Reviewer did not find the arguments for introducing entropy in Section 6. For instance, how do the results of Figure 8 extend the engineering knowledge (since the first buckling mode remains dangerous)? Seemingly, the considered example is improper to illustrate the imperfection problem. In addition, the updated Section 2 must describe the expected improvements to the imperfection modeling. Why must an engineer complicate the issue by considering the entropy if one obtains the same result as the “classical” buckling theory? In this context, the Reviewer wants to mention the statement: “if the first and second critical buckling loads coincide” (Line 396). What is the situation (structure) causing this outcome? How does it affect the imperfection analysis? The wording “The scaling of buckling modes can be used to create an initial imperfection” is also wrong—the bucking mode simulates the imperfections characteristic of a physical structure; the “scaling” procedure is still unspecified. The Author must reconsider the arguments and application examples to make the entropy effect apparent.
  • Section 1 (Introduction) must formulate the initial imperfection problem, define the entropy measure used in this study, and determine the expected improvement to the imperfection analysis. The Authors should remember that the terminology “initial imperfection” is typical for structural engineering but unknown for broad auditory of multidisciplinary journals. Thus, this fundamental issue must be clarified at the beginning of the manuscript. The Reviewer relates the same comment to the word “entropy” (but because of its new nature for engineering science). This section must also cite the relevant literature to substantiate the research program. For example, Figure 1 can confirm nothing. In addition, the Reviewer recommends describing the presentation structure in the last paragraph of the Introduction.
  • To the above comment. Section 3 introduces an elastic strut with pinned supports without any explanation. In other words, such presentation structure makes the impression that these support conditions limit the imperfection analysis. However, this example determines a particular case for illustrating the entropy problem. Therefore, introductory statements or descriptions of the manuscript structure are necessary to introduce the research concept.
  • Section 3.1 requires some introduction. What is the problem considered here? In addition, this section is incomplete—what is the result of these explanations?
  • Section 3.5 is misleading and uninformative. The Author considers two hypothetical cases (Lines 317–326), but the physical interpretation of Tables 1 and 2 is not apparent. For example, the Author can say that the second case is a realistic reflection of the buckling problem, and the scale factor (ci) represents the imperfection measure. Still, the Authors must provide experimental data from the literature [96–98] to substantiate this analogy. Still, the presentation structure on Lines 318–367 is far from logical arguing. Table 1 also looks unnecessary and hiders the essential results of Table 2. The same comment is relevant to Tables 3 and 4. Please do not forget that this study focuses on analyzing the initial imperfections. In addition, what is the physical meaning of the scale factor (imperfection equivalent) belonging to higher buckling modes (i >1)? Please discuss this issue in the manuscript and restructure this essential presentation part.
  • Section 4 is unclear entirely. Is it necessary? If so, please verify the equations, describe all notations, and make a conclusion from this example. For example, the paragraph in Lines 387–389 provides no meaningful information.
  • Section 5 has only one subsection (5.1), which is unacceptable.
  • Abstract reflects all the above problems and must be rewritten. It must reflect the updated presentation structure (i.e., the object, problem, and solution). In addition, the Author must clarify the fundamental terminology, i.e., the initial imperfections and entropy.

2) Equations require a systematic description structure. Partially, this drawback results from the misleading presentation structure (discussed above). Still, it raises the following specific comments:

  • Equation 5. Please check the index “j” ranges specified in the formula and the comment below.
  • Equation 11. The comment “statistical weight” is misleading. Is it necessary? If so, please complete this explanation.
  • Figure 4 (and throughout the text). The piston height, h1, seems constant for all considered cases. Thus, the index “1” becomes misleading.
  • Equation 14 (and throughout the text). Why does the entropy change have two subscripts? In particular, only the first bucking mode meaningfully assesses the internal imperfections. So please either simplify the notations (focusing on the first buckling mode) or clarify the necessity of the highest modes analysis.
  • Equation 16. The wording “proportional to a constant” is misleading. What is the constant mentioned here? What is the physical meaning of this value? The reference “similar to [93]” is unclear. What does it mean? Please also comment on the inequality presented in the formula. What is the piston height assumed in the analysis? The text in Lines 291–292 and 294–296 is uninterpretable.
  • Equation 20 appears in the text without any introduction; the explanation of this formula does not correspond to this equation.
  • Equation 22. The reference “analogous to [93]” is unclear. What does it mean?
  • Equations 24–29. What does the prefix “c” mean?
  • Section 3.5. Please unify the notation of the critical load, which appears as Fcr (instead of Pcr), beginning with this section.

3) Conclusions require modifications to reflect the comment above. In particular, the second paragraph does not clarify the essential contribution of this work to engineering practice. The third paragraph is unclear; the fourth paragraph is unnecessary.

4) Other comments:

  • Line 179. The terminology “beam” typically reflects an element subjected to bending that is incorrect in the considered case.
  • Line 191. The wording “connection with entropy” is misleading. How does it proceed? Improving the presentation structure (Comment 1 above) may solve this issue.
  • Line 243 (and throughout the text). Please avoid the personal writing style.
  • Lines 262–263. The object (gas) must be mentioned regarding the “volume changes.” The terminology “deflection” is misleading in this context (please change it throughout the manuscript); the Reviewer recommends “lateral displacement” or “camber” as appropriate alternatives.
  • Lines 283–284. This comment is misleading. Is the information analogy necessary?
  • The text requires substantial modifications beginning with Line 328.
Comments on the Quality of English Language

The text requires substantial modifications beginning with Line 328. On the one hand, they should reflect Comment 1 above. On the other hand, the majority of explanations require improvements in writing style and clarity.
